# Ketogenic Diet Treatment of Defects in the Mitochondrial Malate Aspartate Shuttle and Pyruvate Carrier

**DOI:** 10.3390/nu14173605

**Published:** 2022-08-31

**Authors:** Bigna K. Bölsterli, Eugen Boltshauser, Luigi Palmieri, Johannes Spenger, Michaela Brunner-Krainz, Felix Distelmaier, Peter Freisinger, Tobias Geis, Andrea L. Gropman, Johannes Häberle, Julia Hentschel, Bruno Jeandidier, Daniela Karall, Boris Keren, Annick Klabunde-Cherwon, Vassiliki Konstantopoulou, Raimund Kottke, Francesco M. Lasorsa, Christine Makowski, Cyril Mignot, Ruth O’Gorman Tuura, Vito Porcelli, René Santer, Kuntal Sen, Katja Steinbrücker, Steffen Syrbe, Matias Wagner, Andreas Ziegler, Thomas Zöggeler, Johannes A. Mayr, Holger Prokisch, Saskia B. Wortmann

**Affiliations:** 1Department of Pediatric Neurology, University Children’s Hospital Zurich, 8032 Zurich, Switzerland; 2Children’s Research Center, University Children’s Hospital Zurich, 8032 Zurich, Switzerland; 3Department of Pediatric Neurology (Emeritus), University Children’s Hospital Zurich, 8032 Zurich, Switzerland; 4Department of Biosciences, Biotechnology and Biopharmaceutics, University of Bari Aldo Moro, 70125 Bari, Italy; 5CNR Institute of Biomembranes, Bioenergetics and Molecular Biotechnologies, 70126 Bari, Italy; 6University Children’s Hospital, Paracelsus Medical University (PMU), 5020 Salzburg, Austria; 7Division of General Pediatrics, Department of Pediatrics and Adolescent Medicine, Medical University of Graz, 8036 Graz, Austria; 8Department of General Pediatrics, Neonatology and Pediatric Cardiology, University Children’s Hospital, Medical Faculty, Heinrich Heine University, 40225 Düsseldorf, Germany; 9Department of Pediatrics, Klinikum Reutlingen, 72764 Reutlingen, Germany; 10University Children′s Hospital Regensburg (KUNO), Hospital St. Hedwig of the Order of St. John, University of Regensburg, 93049 Regensburg, Germany; 11Division of Neurogenetics, Center for Neuroscience and Behavioral Medicine, Children’s National Hospital, Washington, DC 20010, USA; 12Division of Metabolism, University Children’s Hospital Zurich, University of Zurich, 8032 Zurich, Switzerland; 13Institute of Human Genetics, University of Leipzig Hospitals and Clinics, 04103 Leipzig, Germany; 14APHP, Service de Pédiatrie, CHU Jean Verdier, 93140 Bondy, France; 15Clinic for Pediatrics, Division of Inherited Metabolic Disorders, Medical University of Innsbruck, 6020 Innsbruck, Austria; 16Département de Génétique, Unité Fonctionnelle de Génomique du Développement, Hôpital Pitié-Salpêtrière, 75013 Paris, France; 17Division of Paediatric Epileptology, Centre for Paediatrics and Adolescent Medicine, University Hospital Heidelberg, 69120 Heidelberg, Germany; 18Department of Pediatrics and Adolescent Medicine, Medical University of Vienna, 1090 Vienna, Austria; 19Department of Diagnostic Imaging, University Children’s Hospital Zurich, 8032 Zurich, Switzerland; 20Department of Paediatrics, Children’s Hospital Munich Schwabing, MüK and TUM, 80804 Munich, Germany; 21Center for MR Research, University Children’s Hospital Zurich, 8032 Zurich, Switzerland; 22Department of Pediatrics, University Medical Center Eppendorf, 20246 Hamburg, Germany; 23Department of Neuropediatrics, Paracelsus Medical University Hospital Salzburg, 5020 Salzburg, Austria; 24Institute of Human Genetics, School of Medicine, Technical University of Munich, 81675 Munich, Germany; 25Department of Pediatrics, Division of Pediatric Neurology, Developmental Medicine and Social Pediatrics, University Hospital of Munich, Ludwig Maximilians University, 80337 Munich, Germany; 26Institute for Neurogenomics, Computational Health Center, Helmholtz Zentrum München, German Research Center for Health and Environment (GmbH), 85764 Munich, Germany; 27Division of Neuropaediatrics and Inherited Metabolic Diseases, Centre for Paediatrics and Adolescent Medicine, University Hospital Heidelberg, 69120 Heidelberg, Germany; 28Radboud Centre for Mitochondrial Medicine (RCMM), Amalia Children’s Hospital, Radboudumc, 6525 Nijmegen, The Netherlands

**Keywords:** mitochondrial disease, epilepsy, hepatopathy, aspartate glutamate carrier 1 deficiency, AGC1, citrin deficiency, Citrullinemia, treatment, modified Atkins diet, serine

## Abstract

The mitochondrial malate aspartate shuttle system (MAS) maintains the cytosolic NAD+/NADH redox balance, thereby sustaining cytosolic redox-dependent pathways, such as glycolysis and serine biosynthesis. Human disease has been associated with defects in four MAS-proteins (encoded by *MDH1*, *MDH2*, *GOT2*, *SLC25A12*) sharing a neurological/epileptic phenotype, as well as citrin deficiency (*SLC25A13*) with a complex hepatopathic-neuropsychiatric phenotype. Ketogenic diets (KD) are high-fat/low-carbohydrate diets, which decrease glycolysis thus bypassing the mentioned defects. The same holds for mitochondrial pyruvate carrier (MPC) 1 deficiency, which also presents neurological deficits. We here describe 40 (18 previously unreported) subjects with MAS-/MPC1-defects (32 neurological phenotypes, eight citrin deficiency), describe and discuss their phenotypes and genotypes (presenting 12 novel variants), and the efficacy of KD. Of 13 MAS/MPC1-individuals with a neurological phenotype treated with KD, 11 experienced benefits—mainly a striking effect against seizures. Two individuals with citrin deficiency deceased before the correct diagnosis was established, presumably due to high-carbohydrate treatment. Six citrin-deficient individuals received a carbohydrate-restricted/fat-enriched diet and showed normalisation of laboratory values/hepatopathy as well as age-adequate thriving. We conclude that patients with MAS-/MPC1-defects are amenable to dietary intervention and that early (genetic) diagnosis is key for initiation of proper treatment and can even be lifesaving.

## 1. Introduction

Mitochondrial diseases (MDs) are a clinically and genetically very heterogeneous group of inborn metabolic diseases; currently, > 350 mitochondrial disease genes are known [1,2]. Epilepsy is a frequent clinical finding in individuals with MD [3]. No curative treatment for MDs is available, but increasingly, subgroups of patients benefiting from pathomechanism-based supportive treatment (e.g., vitamins and co-factors) are reported [4,5].

### 1.1. The Ketogenic Diet

Ketogenic diet (KD) therapies are isocaloric low-carbohydrate, high-fat diets that shift metabolism towards a reduction of glucose utilisation (glycolysis) and an increase in β-oxidation as well as ketone body production and utilisation. KDs are an established, safe and effective treatment for therapy-refractory epilepsies [6]. Furthermore, the KD is one of the pathomechanism-based treatments for a subgroup of MDs. KD can bypass the pyruvate dehydrogenase complex (PDHc) and is therefore the standard of care in PDHc deficient individuals [7]. There is further evidence that KDs exert their positive effect (among others) via stimulation of mitochondrial biogenesis, improvement of mitochondrial function and decrease of oxidative stress (reviewed in [8]). KDs have been implemented successfully in increasing numbers of MD patients, with and without epilepsy [9].

### 1.2. The Mitochondrial Malate Aspartate Shuttle (MAS)

The mitochondrial malate aspartate shuttle (MAS) (Figure 1) consists of six proteins, of which four are enzymes: cytosolic NAD(H)-dependent malate dehydrogenase (encoded by *MDH1*), aspartate aminotransferase/glutamic oxaloacetic transaminase (*GOT1*) and their mitochondrial counterparts (*MDH2*, *GOT2*). Additionally, two mitochondrial transporters are known: the malate-2-oxoglutarate/malate carrier, OGC (*SLC25A11*) and the two isoforms of the aspartate-glutamate carrier, AGC1 (also known as aralar; *SLC25A12*) mainly expressed in the brain and AGC2 (also known as citrin; *SLC25A13*) in the liver. These proteins enable electron transport from the cytosol into the mitochondria, as the inner mitochondrial membrane is impermeable for the electron carrier NADH. By shuttling NADH across the mitochondrial membrane in the form of a reduced metabolite (malate), the MAS plays an important role in mitochondrial respiration, when glucose or lactate are the substrates. In addition, the MAS maintains the cytosolic NAD^+^/NADH redox balance, by using redox reactions for the transfer of electrons. This explains the importance of the MAS in sustaining cytosolic redox-dependent metabolic pathways, such as glycolysis and serine biosynthesis (Figure 1). 

Human disease has been linked to biallelic pathogenic variants in *MDH1*, *MDH2, GOT2* and *SLC25A12* (Figure 1) sharing a clinical overlapping neurological phenotype with epilepsy as dominating feature [10]. Further, biallelic pathogenic variants in *SLC25A13* are known to cause the complex disorder citrin deficiency causing a spectrum of phenotypes that are highly age-dependent (neonatal intrahepatic cholestasis caused by citrin deficiency (NICCD), failure to thrive and dyslipidemia caused by citrin deficiency (FTTDCD), and adult-onset recurrent hyperammonemia with neuropsychiatric symptoms in citrullinemia type II (CTLN2)) [11,12]. Citrin deficiency is a more prevalent condition in Asian populations, particularly in Japan [13] but anecdotal reports also describe Caucasian patients [14]. Citrin deficiency is often also referred to as citrullinemia type II in analogy to citrullinemia type I (argininosuccinic acid synthetase deficiency), the latter being a classical urea cycle disorder requiring a dietary treatment with high carbohydrates but low protein. This high carbohydrate supplementation would also be the ‘intuitive’ treatment for hypoglycaemia, one of the features often seen in the ‘NICCD’ and ’FTTDCD’ subtypes. However, citrin deficiency is one of the very rare contraindications among the inborn metabolic diseases for a high carbohydrate (emergency) regime.

It is further important to realize that ‘NICCD’ is generally not severe, although liver transplantation has been required in rare instances. With appropriate treatment symptoms often resolve by age of one year. However, after a ‘silent’ period of up to two decades, individuals with a history of ‘NICCD’/’FTTDCD’ can proceed to ‘CTLN2′ with risk of deterioration upon e.g. medication, alcohol, surgery or high carbohydrate intake.

Somatic biallelic variants in *SLC25A11* have been found in paragangliomas and pheo-chromocytomas [15].

### 1.3. The Mitochondrial Pyruvate Carriers 1 and 2

The mitochondrial pyruvate carriers (MPC) 1 and 2 (*MPC1, MPC2*) transport pyruvate, which is mainly generated by cytosolic glycolysis, to the mitochondrial matrix (Figure 1). There, it is one of the main substrates provided to the tricarboxylic acid (TCA) cycle and the respiratory chain. This transporter plays a key role in glucose catabolism as well as mitochondrial respiration and the metabolic changes—as well as the possibility of bypassing via KD—are comparable with the ones in PDHc deficiency. Human disease has been related to deficiency of MPC1, to date not to MPC2.

### 1.4. Ketogenic Diet for MAS-Defects and MPC1-Deficiency in the Literature

Within the MDs, the subgroup of disorders related to the MAS as well as to the MPCs are ideal candidates for a pathomechanism-based dietary treatment. The MAS is driven by the NADH/NAD+ ratio. Low-carbohydrate and high-fat diets do not only reduce glycolysis, thereby decreasing the NADH/NAD+ ratio and thus favouring cytosolic aspartate production by the cytosolic part of the MAS (aspartate amino transferase) but additionally provide ketone bodies and fatty acids as alternative substrates for the TCA cycle which provides NADH and FADH2 to the respiratory chain (Figure 1) [10]. A fat-enriched and carbohydrate-restricted (normal protein) diet is the therapy for *SLC25A13* related citrin deficiency. It is important to note, that this is not a classical ketogenic diet as no ketosis is aimed for, but rather the intention to reduce glucose. KD was further reported to be effective in two individuals with *SLC25A12* related AGC1-deficiency, improving epilepsy in both [16,17]. In one, also the myelination in magnetic resonance imaging (MRI) was followed up and improved [16]. Additionally, intraperitoneal treatment with the ketone body β-hydroxybutyrate (BHB) was shown beneficial in a mouse model [18].

In contrast, KD has been reported unsuccessful in one case with MPC1-deficiency [19]. An interesting observation was made in a mouse model, where embryonic lethality was rescued by a KD given to the unaffected pregnant dams [20]. However, their affected offspring died postnatally, irrespectively of the KD.

Taken together, KD is a promising pathomechanism-based dietary treatment approach for the disorders of the MAS as well as MPC1-deficiency. We here investigate a cohort of 40 affected individuals regarding the phenotype, genotype and the course on KD.

## 2. Materials and Methods

In this retrospective study, individuals were recruited from the GENOMIT/PerMim collaborative research project. Inclusion criteria were biallelic variants in *MDH1, MDH2, GOT2, SLC25A12*, *SLC25A13* or *MPC1* classified as (likely) pathogenic according to the American College of Medical Genetics and Genomics (ACMG, Bethesda, USA) guidelines [21]. Further eligible individuals were identified via literature review (PubMed search with restriction to the English language completed October 2021). If individuals were published previously, the respective authors were contacted for an update. If this was not received, only published data were included. All (de-identified) data were retrieved retrospectively via case report forms. Additionally, a written case report, MRI and magnetic resonance spectroscopies (MRS) were requested. All available MRIs were reviewed by the same investigator (E.B.). The MRS acquisition and analysis in the individual AGC1-1 is detailed in the Appendix A.

No literature review for *SLC25A13* was performed as the treatment is well established and the data and their analysis would go beyond the scope of this article (for review see [11,22]). 

Functional characterization of the AGC1 variants in patients AGC1-2/3 by measuring transport capacity and for AGC1-5, western blot analysis was performed in patient-derived fibroblasts (see Appendix A).

## 3. Results

A summary of the clinical findings is presented in Table 1 (a–d) and Table 2 and Figure 2, the genetic findings in Table 2 and Table 3, and the follow-up on KD in Table 4. The detailed written case reports can be found in the Appendix A. 

From the defects with a predominantly neurological phenotype, we identified 14 AGC1/*SLC25A12*-deficient individuals of whom eight were treated with KD. For the other defects, the subject numbers were as follows (defect, (total/treated)): MDH1 deficiency (2/0), MDH2 deficiency (4/3), GOT2 deficiency (4/0), MPC1 deficiency (8/2).

For the hepatic phenotype of citrin/*SLC25A13*-deficiency, we identified eight individuals, of whom six followed a carbohydrate-restricted and fat-enriched diet.

Throughout the following text, specific features will be reported for the X subjects of the Y subjects with available information on that specific feature as (X/Y).

### 3.1. AGC1-Deficiency (SLC25A12)

#### 3.1.1. Presentation before Initiation of KD

Fourteen affected individuals from 12 families were identified (Table 1a), of whom six from five families were previously unreported (AGC1-1 to AGC1-6) [16,17,23,24,25,26,27,28]. The median age at presentation was two weeks (range: neonatal period–seven months) and the last follow-up (reported in 12/14) at a median of 5 years 7 months (range: 13 months–11 years); AGC1-13 died at the age of seven years.

The phenotype (Figure 2) was homogenous, with early onset epilepsy (13/13) and profound to severe global developmental disability (GDD) (12/12). Muscular hypotonia/weakness was seen in 10/13 individuals, of whom three developed spasticity. Two out of nine individuals had a primary developmental impairment, and all (9/9) showed regression or stagnation upon seizure onset. All individuals remained non-verbal (12/12). Highest achieved motor milestones reported were the ability to bear weight and to sit independently, in one case (AGC1-9) with the last follow-up at 13 months. Hyperkinetic movement disorder was reported in 4/9 individuals. 7/10 individuals had microcephaly, reported as secondary in four of these cases. Failure to thrive (FTT) and/or feeding difficulties were reported in 5/7 cases. 

Age at epilepsy onset was reported in 13/14 and was between 40 days and 10 months (median 7 months). Epilepsy was refractory to therapy in 7/11 individuals. One subject (AGC1-12, [26]) was seizure-free at the age of three years when treated with phenobarbital and levetiracetam and later even without anti-seizure medication (ASM) up to the age of 12 years. Then, seizures relapsed and were controlled with levetiracetam again. The remaining three individuals (AGC1-2/3/9) had only one ASM tried but did not become seizure-free. Description of seizures/epilepsy was available in 10 individuals. Seizures were focal in origin with variable semiology (apnoea (4/10), tonic (3/10), clonic (4/10), tonic-clonic (3/10), myoclonic (2/10), seizures with odd laughter (2/10)); clusters of seizures (1/10), or status epilepticus (4/10) were reported as well.

The electroencephalogram (EEG) was abnormal in all (10/10) with diffuse background slowing (7/10), focal slowing (2/10) and focal/multifocal or unspecified epileptic discharges (or seizure patterns) (9/10). In the remaining case (1/10) a voltage attenuation was the only feature reported.

Laboratory investigations revealed increased serum lactate levels (max. 7.2 (<2 mmol/L)), at least intermittently, in 9/12 individuals. Cerebrospinal fluid (CSF) lactate was elevated in 3/4 individuals (max. 3 (1.1–1.7 mmol/L)). In AGC1-1, CSF amino acids were analysed repeatedly. Serine was low normal to reduced (1.5–22.6 (21–44 μmol/L)) and normalised (27.5–58.3 μmol/L) after supplementation with serine and glycine (before the introduction of KD). Aspartate was < 2 μmol/L to undetectable (2–4 μmol/L). Levels of these amino acids in plasma were within normal range.

Full MRIs were available in 3/14 individuals, and descriptions or exemplary images of MRI in an additional 7/14 individuals with one to five studies per subject. In two subjects, neuroimaging on KD was available with one and two MRIs, respectively. Neuroimaging was best documented in AGC1-1, with three consecutive MRIs (at age four months, seven months, and 21 months) before and one follow-up MRI 19 months after initiation of KD. Representative images are shown in Figure 3. Two major features were found in all (untreated) cases (10/10): (1) progressive cerebral volume loss as evident from enlargement of the third and the lateral ventricles as well as dilated extracerebral spaces, (2) extensive abnormal signal characteristics in the supratentorial white matter showing a hyperintense signal in T2w as seen in “secondary (to neuronal disease) hypomyelination” [32] associated with thinned corpus callosum. In 2/10 subjects, multiple MRIs revealed progression of myelination with age without becoming age appropriate. Normal myelination for age was reported in 2/10 early in the course (at seven, 10 months). In 3/10 individuals, cerebellar atrophy occurred and abnormalities in deep grey matter structures were reported in 2/10 individuals. 

Eleven MRS were performed in 6/14 individuals, between the age of four months and six years (median 21 months). In AGC1-1 and AGC1-7, MRS was performed in parallel with each MRI (Figure 4, Table 5, [16]). Before treatment, the main findings were a reduced N-acetyl-aspartate (NAA) peak in 10/10 studies of 5/5 subjects and an increased myoinositol (mI) peak in 7/7 studies in 5/5 individuals (in AGC1-12, an MRS was done but neither NAA nor mI was reported). Lactate peak was reported in four individuals (in seven MRS) and was increased in five MRS of 3/4 subjects and normal in two MRS of 2/4. AGC1-1 showed initially normal and later, elevated lactate (Table 5).

#### 3.1.2. Follow Up during KD

Eight of 14 individuals [16,17] received a classical KD (cKD), the most restrictive form of KD with a fixed ratio (2:1 to 4:1) between fat and non-fat nutrients (median age at initiation of KD was 2.2 years (14 months–6 years), median duration 19 months (1 month–2 years 7 months)). Ketosis was at a therapeutic level (BHB > 2 mmol/L) in 4/4 individuals. 

Five out of eight individuals were still on cKD at the last follow-up. In AGC1-4, in whom cKD was initiated because of therapy refractory epilepsy when the genetic diagnosis was still unknown, cKD was stopped after 1 month because of intolerance (vomiting and gastrointestinal problems) and lack of efficacy. However, ketosis was not reported in this subject. The other two, the siblings AGC1-2 and AGC1-3, were discontinued because the cKD was not practicable anymore for the family.

The most evident effect of cKD was a rapid and dramatic reduction in seizure frequency in 7/8, with seizure freedom at the last follow-up in all (5/5) individuals, who were still on cKD. Seizures relapsed associated with low ketosis (accidental or due to non-compliance) in 2/8, and in 1/8 during therapeutic ketosis. However, in these individuals, an increase in the ketogenic ratio from 3:1 to 4:1 led to seizure freedom again. ASM could be tapered off in 2/6 subjects with efficacy of KD, were reduced in 1/6, and remained unchanged in 3/6. The EEG findings improved in 6/6 individuals and normalised in one of these. 

Muscle tone, head control and development improved considerably in 6/8 and remained unchanged in the other subjects. Spasticity developed in 2/6 and AGC1-2 already had limb spasticity before the introduction of cKD at age 5 years 8 months and had no benefit from cKD on muscle tone nor development. His younger sister though, who started cKD at the age of 1 year 4 months, was (still) hypotonic and improved on cKD.

The highest motor ability (AGC1-5) was independent walking, achieved within two years on diet at age 4 years 6 months. This individual was neither sitting nor crawling before the introduction of cKD. 3/8 individuals learned to sit independently. Social interaction and communication were reported to improve considerably in 4/8 subjects but all remained non-verbal.

The choreo-athetotic movements in AGC1-1 ceased almost immediately after the introduction of cKD, and hypotonic-dyskinetic movement disorder in AGC1-5 improved, enabling him to achieve independent walking.

2/7 of individuals who initially showed impressive developmental progress, experienced stagnation of development. Of those, ketosis was stable in one (AGC1-1) and highly fluctuating due to probable mal-compliance in the other (AGC1-3).

Laboratory follow-up was only available for AGC1-1 and AGC1-3. In AGC1-1, abnormal liver function tests just before the introduction of cKD (gamma-glutamyl-transferase (gGT) 694 (< 23 U/L), alanine amino transferase (ALT) 88 (< 28 U/L), aspartate amino transferase (AST) 47 (< 50 U/L) normalised within six months. CSF analysis was not repeated. In AGC1-3, serum lactate levels decreased from 3.1–5.7 mmol/L to 1.6–3.5 mmol/L. 

MRI follow-up after the introduction of cKD was available in 2/8 cases. AGC1-1 had one imaging study after 19 months, AGC1-7 two (after six months and 19 months). A substantial reduction in the size of the ventricles and extracerebral spaces and a marked improvement in myelination was seen (Figure 3, [16]).

MRS follow-up was available in all three imaging studies of the 2/8 individuals with MRS before cKD (for AGC1-1 see also Table 5 and Figure 4). The NAA concentration and ratio to creatine increased from pre-cKD values in AGC1-1. In AGC1-7, a comparison between MRS after six months to pre-treatment MRS was only possible for NAA/creatinine ratio, which improved. NAA concentration after six months was normal in basal ganglia but still reduced in occipital white and grey matter. After 19 months, NAA concentration increased in white matter and grey matter compared to the first MRS on treatment, the NAA/creatinine ratio remained stable. In AGC1-1, mI and lactate normalised, choline remained normal compared to the MRS just before the introduction of cKD at 21 months of age. Information on these compounds was lacking for subject AGC1-7.

### 3.2. MDH1-Deficiency (MDH1)

The literature search revealed two children (cousins) with MDH1-deficiency ([29], Table 1b). None of them was treated with KD. The initial presentation was described in MDH1-1 only. He was born at 32 weeks of gestation and in NICU (7 days) for weight management. One week after discharge he was re-hospitalized because of respiratory distress. When published, they were 2 ½ (MDH1-1) and 4 years (MDH1-2) old, though clinical features of MDH1-1 were updated at the age of 14 months at the latest. In both, the central nervous system was predominantly affected by global developmental delay, epilepsy and strabismus. Microcephaly was reported in both, as progressive in one. Both had comparable dysmorphic features: plagiocephaly, bulbous nose, deep eyes, frontal bossing and micrognathia. Laboratory investigations showed normal plasma amino acids, acylcarnitine, lactate, and urine organic acid analysis. In dry blood, high glutamate and reduced glutamine/glutamate ratio were found and untargeted metabolomics revealed increased glycerol-3-phosphate (G3P). Cerebral MRI showed abnormalities in the corpus callosum in both. Additionally, in MDH1-1, reduced myelin in posterior regions, cerebral atrophy and mild hypoplastic inferior vermis and pontine hypoplasia was reported.

### 3.3. MDH2-Deficiency (MDH2)

#### 3.3.1. Presentation before Initiation of KD

There are four unrelated subjects with MDH2 deficiency published to date ([30,31], Table 1b). The initial presentation was between birth and five months with muscle hypotonia (2/4), severe constipation (1/4), febrile seizures (1/4) and macrosomia, macrocephaly and two supernumerary nipples (1/4). Three subjects were still alive at the last follow-up at four, five and 12 years, respectively. MDH2-2 died at the age of 1 years 8 months secondary to metabolic decompensation.

All subjects developed epilepsy in the first months of life, which became refractory to ASM in 3/4. Developmental delay and muscle hypotonia were reported in all and muscle weakness in 3/3. Two out of three were reported to have dystonia and 1/3 additional dyskinesia. Pyramidal signs were described in 2/3. MDH2-4 experienced a metabolic stroke in the left nucleus lentiformis, causing hyperkinetic and choreatiform movement disorder. The highest achieved milestones were crawling at 18 months (MDH2-1) and babbling at 12 months (MDH2-2). Strabismus was reported in 2/2, whereas in one of them, a retinitis pigmentosa was diagnosed at the age of 4 years (MDH2-2). Failure to thrive necessitating tube feeding was seen in 3/4, while 1/3 showed body measurements within the upper normal limits (MDH2-2 at 18 months +1SD for HC, length and weight). 

All subjects showed elevated plasma lactate (2.8–5.7 (<1.7 mmol/L)), lactate to pyruvate ratio (20–63 (<18)) and 3/3 had increased CSF lactate (2.25 and 3.3 (<1.7 mmol/L)). Elevated urinary malate (2/2) and fumarate (in 2/3, at least in some measurements) was found, while succinate was normal in 3/3. 

Brain MRI showed cerebral atrophy in 3/3, most prominent in the anterior parts of the brain in 2/3, cerebellar atrophy in 2/4 and delayed myelination in 3/4. Germinolytic (pseudo)cysts were described in 1/4. 

MRS was done in 3/4 and revealed elevated lactate peak in all of them.

#### 3.3.2. Follow Up during KD

cKD was given to 3/4 subjects (Table 4). A reduction of seizure frequency was reported in 2/3 subjects. No further information about the diet and its effect is available. However, 1/3 individuals died shortly after the introduction of KD (1/2 subjects with benefit on epilepsy). He had vomiting, coughing, tachypnoea and impaired consciousness associated with fatal metabolic decompensation (lactate acidosis, excretion of ketone bodies and TCA cycle metabolites). MDH2-4 received triheptanoin and had improvement in muscle tone, motor abilities and communication [31].

### 3.4. GOT2-Deficiency (GOT2)

Four affected individuals from three unrelated families have been reported ([32], Table 1c). None of them was treated with KD. All cases had mainly central nervous system (CNS) affected and presented within the first month of life with feeding difficulties (4/4), drooling (2/4) and developmental impairment (4/4). The last clinical follow-up was between four and 10 years of age. 

Maximal motor achievement was the ability to sit independently (GOT2-3 at eight years) and the best language skill was speaking three to four words (GOT2-2 at 10 years). All (4/4) subjects had muscle hypotonia initially, and later developed spasticity and progressive microcephaly. Infantile epilepsy (onset at age 4 to 9 months) occurred in 4/4 and became refractory to ASM in 2/4. All individuals were prone to infections. 

Laboratory investigations showed increased serum lactate in 4/4 (3.0–5.7 (0.5–2.2 mmol/L)) and hyperammonaemia in 4/4 (during infancy: between 110 and 143 (16–60 μmol/L) in 4/4). For the two individuals not treated with serine and pyridoxine, the values were still increased at the age of 10 (GOT2-2) and eight years (GOT2-3) (70 and 75 (11–32 μmol/L)). Reduced plasma serine (47 (70–294 μmol/L)) and increased citrulline (9 (7–55 μmol/L)) were measured in 1/3 (GOT2-1) at age 14 months. CSF amino acid levels were not reported. 

Brain MRI showed cerebral atrophy in all individuals, a hypoplastic vermis in 3/4 and a thin (2/4) or hypoplastic (1/4) corpus callosum. 1/4 had additional multi-cystic encephalomalacia. No follow-up MRIs were reported.

#### Treatment

Treatment with serine and pyridoxine was given in variable dosages in 2/4 individuals (GOT2-2, GOT2-3). On this treatment, both individuals that had refractory epilepsy before and achieved seizure freedom, and in ½, it was possible to taper off all ASM. 2/2 were described as more alert. 1/2 started to fixate objects. Intellectual disability at last follow-up (on treatment) was profound in both.

### 3.5. MPC1-Deficiency (MPC1)

#### 3.5.1. Presentation before Initiation of KD

Eight individuals with MPC1 deficiency from six unrelated families were identified ([33], Table 1d). Three families have not been reported yet (MPC1-1 to 4). Additionally, in one family MPC1-deficiency was diagnosed metabolically in a consecutive pregnancy, which was terminated [19]. 

The initial presentation was—if reported—at birth in 4/5 with muscle hypotonia in 3/4, respiratory distress and elevated lactate in 1/5. Seizures were the first symptom in 1/5 (onset 6 years). One child deteriorated neurologically and deceased at the age of 1 year 7 months (MPC1-5). The latest follow-up in the others was between five and 20 years.

The clinical presentation was variable regarding the involved organs and the severity of the disease. However, all 8/8 had CNS involvement and developmental impairment. Severity was variable even within a family (MPC1-7/8). In 3/8, the developmental impairment was mild. In one of them, initial milestones were achieved normally but, at the age of 12 years, his IQ was tested 56. Developmental impairment was severe to profound in the remaining individuals. Other CNS features were muscle hypotonia (5/6), epilepsy (4/5) and microcephaly (3/5). The epilepsy phenotype was reported in 3/4. Subject MPC1-1 had myoclonic seizures at 2 ½ months and infantile spasms from eight months on. MPC1-3 had treatment-responsive tonic and tonic-clonic seizures from four months on and MPC1-2 had bilateral tonic-clonic seizures from six years on. 

Features affecting other than CNS were reported in 4/6 subjects. Peripheral neuropathy was seen in the siblings MCP1-7/8, and visual impairment in MPC1-7. Individual MPC1-2 had splenomegaly, insulin-dependent diabetes (antibody negative) and multiple fractures after minor trauma. Subject MPC1-5 presented with facial dysmorphism, hepatomegaly, congenital heart defect and renal insufficiency. Growth retardation was reported in two. For the previously published subjects MPC1-6 to 8, the age at introduction of cKD was not reported. The clinical features were included here, even though we presume that at least part of them occurred while on cKD.

Before the introduction of cKD, individual MPC1-1 was very sleepy and was not able to change his position, nor grasp objects and had almost no eye contact, no speech and treatment-refractory infantile spasms (eight ASM used before).

Serum lactate was increased in all 6/6 subjects and serum pyruvate in 5/5. The lactate/pyruvate ratio was reported in two individuals as normal to low (5–10 (9–18)). Urine lactate and pyruvate were increased in 4/5. In MPC1-1, a neonatal CSF examination showed increased lactate (5.8 (1.1–2.6 mmol/L)), while it was 6 mmol/L in serum one hour postpartum. Glucose CSF to blood ratio was decreased by 0.41 (normal > 0.55).

The original MRI was only available and reviewed from subject MPC1-1. Reports on six MRIs in five subjects were available. Common features in the MRIs with abnormalities (3/6) were cerebral atrophy (2/6) and reduced myelin (2/6). In addition to cerebral atrophy, subject MPC1-5 had periventricular leukomalacia and calcifications. The detailed visual evaluation of the second MRI of MPC1-1 at 13 months showed atrophy with a reduction of supratentorial white matter, an e vacuo dilation of inner and outer CSF spaces, symmetric signal alterations of pallidum and medial thalami with corresponding impaired diffusion.

MRS showed no abnormalities in 2/3 and increased lactate peak in the caudate nucleus in 1/3.

#### 3.5.2. Follow-Up during KD

Treatment with KD was initiated in 5/8 individuals with follow-up information in 2/5 (MPC1-1 and MPC1-5), hence only these two were regarded as treated for further evaluation (Table 4).

MPC1-5 showed no improvement on cKD, which was started in the neonatal period. BHB was sub-therapeutic before the age of 16 months. The only therapeutic level reported was 3.7 mmol/L after a meal at 16 months. Therefore, in retrospect, the KD might have been insufficiently strict to be successful. She had a progressive neurological disease and died at 1 year 7 months of age. In contrast, individual MPC1-1 had distinct improvement in his clinical condition. The last follow-up on cKD was after four years of treatment (age 5 years 3 months). The cKD had positive effects on development and neurological state. The child was more alert, social interaction was better, he had better eye contact and he recognized people. Muscle tone and head control increased, he was able to roll over and was trying to sit up. However, the subjects’ secondary microcephaly was progressive. At the start of cKD, spasms were treated with vigabatrin and lamotrigine. On cKD, an initial reduction of spasms was seen after one week. Spasms increased after 2 ½ months again and exacerbated after six months. At that time, topiramate was introduced and cKD was intensified from 2.5:1 to 4:1. This led to a seizure-free period of 12 months, even though vigabatrin and topiramate were tapered off. Then, myoclonic seizures occurred. The increase of lamotrigine reduced seizure frequency and at the latest one to two seizures per day were reported.

The serum lactate levels in MPC1-1 normalised during KD.

No follow-up MRIs on KD were reported.

### 3.6. Citrin Deficiency (Also Known as AGC2-Deficiency, SLC25A13)

The eight previously unreported individuals AGC2-1 to AGC2-8 of Caucasian origin are summarized in Table 2 and detailed in the Appendix A. They represent the highly age-dependent spectrum of diseases associated with citrin deficiency. Of note, four were born SGA, and for two others, this information was unavailable.

Two individuals died at first presentation, and the diagnosis was made only post-mortem. Their laboratory results mimicked a urea cycle disorder (AGC2-2 at age 35 years, ‘CTLN2-subtype’) and tyrosinemia (AGC2-5 at age seven months), respectively, which led to subsequent treatment with high carbohydrates and protein restriction, resulting in death. 

The other six individuals received the correct diagnosis and an adequate diet with carbohydrate restriction and fat enrichment. This led to improvement and (near) normalisation of laboratory parameters as well as adequate thriving in all of them. Despite this, signs of fatty liver disease were present in three of them. Of note, none had shown hypoglycaemias.

Individuals AGC2-3 and AGC2-8 with ‘NICCD-subtypes’ were detected by new born screening and treated from the age of five and four weeks (both currently aged 4 ½ years), respectively. AGC2-7 represents the ‘FTTDCD-subtype’ with failure to thrive, cholestasis and elevated triglycerides recognized at the age of six months. AGC2-4 and AGC2-6 were diagnosed based on their positive family history aged four weeks and 12 days, respectively. In line with their early diagnosis, they had only mild laboratory anomalies that normalised within seven to 10 days. The remaining subject (AGC2-1) presented aged four months with hepatomegaly and suggestive laboratory findings. Of note, none of the individuals with ‘NICCD’ or ‘FTTDCD’-subtypes had shown hypoglycaemias.

### 3.7. Genotypes

As shown in Table 3, we identified 34 different variants in our cohort, of these, twelve variants have not been published before and extend the genotypic spectrum. Of note, in our Caucasian individuals with citrin deficiency two variants previously reported in Asian individuals were found and four novel variants.

### 3.8. Functional Investigations

The variant c.1618G > A identified in AGC1-2/3 causes the substitution of the amino acid aspartate in position 540 into an asparagine. The aspartate in position 540 is a part of the sequence motif PX[D/E]XX[K/R] conserved in all mitochondrial carriers [38] and is involved in the salt-bridge network on the matrix side that is important for the function of the mitochondrial carriers [39]. To test the functional relevance of the identified mutation we measured the transport activity of the D540N mutant upon reconstitution of purified recombinant proteins into liposomes. As shown in Figure 5A, the substitution of Asp540 with asparagine caused a complete loss of activity in accordance with the predicted involvement of Asp540 in the transport mechanism. 

In AGC1-5 the c.225del mutation generates an early truncated and not functioning form of the protein, p.(Glu76Serfs*17); the second mutation (c.1747C > A) is a synonymous variant predicted to alter the splicing. Therefore, we performed western blot analysis on fibroblasts confirming that compound-heterozygous variants led to a complete loss of the AGC1 protein expression in the subject (Figure 5B).

For detailed information on functional investigations see Appendix A.

## 4. Discussion

By collecting and analysing this cohort of a total of 40 individuals with defects in the MAS/MPC1, we can further delineate the phenotypes (Figure 2, Table 1 and Table 2) and extend the genotypic spectrum with 12 novel variants. Individuals with **defects in the MAS** showed a neurological phenotype in all defects. The complex disorder citrin deficiency can present with an isolated hepatic clinical presentation in infancy and childhood but also lead to a neuropsychiatric phenotype in adulthood. In the thirteen of the “neurological MAS” individuals, who were treated with KD, we outline the clinical course and add further evidence for the efficacy (Table 4).

### 4.1. Phenotype

This “**neurological MAS**” phenotype was uniform with early onset epilepsy, muscle hypotonia and a severe global developmental impairment with reduced cerebral myelin. The shared but non-specific metabolic marker was—at least intermittently—elevated lactate. This is in line with the predominant cerebral expression and the function of the MAS: linking glycolysis with mitochondrial respiration as well as the synthesis of aspartate and glutamate, the most important excitatory neurotransmitter (reviewed in [10]).

Paucity of cerebral myelin was seen in all neuronal MAS-defects but GOT2-deficiency. In AGC1-deficiency, the reduction of normal myelin signal was interpreted as a secondary phenomenon, as a consequence of neuronal dysfunction [41,42]. Not only AGC1, but all MAS components play a crucial role in mitochondrial respiration, when glucose or lactate are the substrates. A defect of this shuttle leads to a shortage of energy supply leading to a malfunction or degeneration of neurons. Intact neuronal metabolism is necessary for the maintenance and formation of myelin. Another proposed mechanism leading to reduced myelin is the importance of MAS in supplying precursors of myelin synthesis such as aspartate and NAA [16,43]. A low level of aspartate and NAA, together with reduced myelin, was found in the brains and cultured neurons of AGC1 knock-out (KO) mice [43]. In agreement with this, aspartate was not detected in CSF of AGC1-1, a finding reported for the first time in humans. Interestingly, aspartate was normal in serum, likely because AGC2 is the main isoenzyme expressed in the liver. Unfortunately, no CSF on KD was investigated. However, aspartate increased from pre-KD MRS to MRS on KD (data not shown). A drawback is that no reference values for aspartate in MRS are available. Furthermore, aspartate serves as amino donor for de novo glutamate synthesis [44]. Neurotransmitter imbalances are well-known reasons for the development of seizures and epilepsy, a main clinical feature of these individuals.

The subjects with **MPC1-deficiency** showed a broader phenotypic spectrum. Also, in MPC1-deficiency, glycolysis-related energy production, the main energy source for the brain is hampered [45]. All individuals had developmental impairment but of very variable severity, and epilepsy was only reported in less than half of the individuals, with a greater age range of onset. MPC1 is expressed in many organs, and we here present the first individual with the additional features of splenomegaly, bone fractures after minor trauma and insulin-dependent, anti-body negative diabetes mellitus. This is in line with findings in a mouse model with hypo-morphism of Mpc2 which showed impaired glucose-stimulated insulin secretion [46]. Additionally, the inhibition of MPC was shown to play a key role in the regulation of insulin secretion in a cellular model (832/13 β-cells) and isolated pancreatic islets from rats and humans [47] and a general role of MPC in diabetes is outlined in [48]. Other clinical features were peripheral neuropathy, muscle weakness, dysmorphic features, congenital heart defect and growth failure [19,33].

### 4.2. Effect of KD on Neurological MAS-Defects and MPC1-Deficiency

Of the 13 KD-treated individuals with the neuronal MAS-defects and MPC1-defect, the KD was mainly beneficial in 11, at least temporarily. Ten of the 13 (77%) subjects had substantial seizure reduction and five with AGC1-deficiency were still seizure free at the last follow-up. This indicates a higher efficacy of KD in this cohort than the generally expectable 40–50% chance of at least 50% seizure reduction [49]. The majority remained on the KD, not only pointing to a high efficacy but also tolerability.

In AGC1- and MDH2-deficiency, the most striking and most immediate effect was the one against seizures in these severe epilepsies. A reason might be that KD acts not only by bypassing the defect but has also a general antiseizure effect [49]. Though much less impressively, also muscle tone and motor abilities improved. The individuals were more alert, but communicative skills remained very basal and developmental impairment was still severe to profound. The data we collected on previously unreported cases were in agreement with the previously published data. In MPC1-deficiency, we report for the first time a beneficial clinical course on KD. Additionally, the efficacy against epilepsy was paramount, while, besides more alertness and social interaction, no clear-cut improvement of development was seen.

In addition, we can show that the liver dysfunction in one individual (AGC1-1) improved rapidly and normalised about 6 months after the introduction of KD, which might be a hint that the liver dysfunction was a direct consequence of the metabolic defect. We have no reports on hepatic transaminases from other subjects.

Follow-up cerebral MRI on KD was only available in two subjects with AGC1-deficiency The previously published findings of AGC1-7 [16,23] were confirmed in the study of AGC1-1, namely a striking increase in myelin and a decrease of inner and outer CSF spaces indicating an increase of volume (Figure 3). All these changes point towards a recuperation of cerebral neurons.

Follow-up MRS on KD was performed in the subjects with AGC1-deficiency in whom MRI was done, too. The findings in the individual published by Dahlin et al. [16] were confirmed, in that in AGC1-1 an increase in NAA was also seen (see Table 5) in both NAA/creatinine ratio and NAA concentration. It is not completely known, why NAA is reduced in AGC1-deficiency. One reason for low NAA in AGC1-deficiency was proposed to be diminished NAA synthesis as a consequence of the low precursor aspartate (see Figure 1, [16,43]). Further, NAA is a surrogate marker of neuronal density and integrity [50] and decreases secondary to neuronal degeneration.

The analysis of the MRS in the subject AGC1-1 showed spectral peaks in the residuals of the LCModel which did not fit with any metabolites attributed in the basis set [51]. Therefore, the model was complemented by simulated peaks for glycerol, which led to a reduction of these residuals, or in other words showed a good fit with the spectra (see Figure 4). The peaks thus reflect glycerol or a very similar molecule. In a second step, MRS was run on a glycerol-3-phosphate (G3P) phantom which revealed comparable peaks. These Glycerol/G3P peaks were seen only before treatment with KD and resolved on KD. This points to an increase in G3P in AGC1. In AGC1-KO mice, increased activity of the G3P-shuttle was suggested. This G3P-shuttle, mainly situated in astrocytes, recycles NAD^+^ and is thus an alternative to lactate production [10,52] (see Figure 1). Because the rate-limiting step in the G3P-shuttle is the isomerisation from G3P into dihydroxyacetone phosphate, G3P would accumulate. Thus, our findings support the suggestion of increased G3P-shuttle activity and the rescue of the MAS defect in AGC1-deficiency by KD. Furthermore, there exists also evidence for increased G3P in another MAS defect, the MDH1 [29].

### 4.3. Serine Supplementation

Another consequence of MAS defects is reduced serine. One mechanism might be reduced serine-synthesis, a metabolic branch of glycolysis, which is also dependent on the availability of NAD^+^ (see Figure 1). However, in the brain, serine is mainly synthesised by glial cells (for a review [53]). Therefore, another explanation for low serine, proposed by Pardo et al. [44], would be increased serine-utilization for pyruvate-synthesis, as the pyruvate-lactate ratio drops. Reduced serine was previously described in GOT2-deficiency [32] and low CSF-serine was reported for the first time in one of the AGC1-deficient individuals (AGC1-1). In GOT2-deficiency, serine was supplemented together with pyridoxin/vitamin B6 because GOT2 is pyridoxin-dependent. And a clinical improvement was observed. No KD was performed. In the individual with AGC1-deficiency and low CSF-serine, serine was substituted (together with glycine) and normalised, but no clear clinical benefit was seen. In this subject, it was the treatment with KD that led to an impressive clinical and MRI/MRS improvement. This indicates that the reduced CSF serine is not the main cause of the clinical features and serine supplementation does not rescue the NADH/NAD imbalance caused by AGC1 deficiency. However, supplementation still might be supportive.

Taken together, this cohort highlights once again the need for early genetic diagnostics in children with epileptic seizures and developmental delays.

### 4.4. Citrin Deficiency

In line with this, we wish to raise awareness of the complex disorder of citrin deficiency in Europe. Our eight cases of citrin deficiency were all but one (German case) diagnosed and treated in Austria. Due to confusing laboratory markers—and presumably additionally because of the rarity of citrin deficiency outside the Asian population—citrin deficiency was not suspected in two cases and treatment with high carbohydrates and protein restriction was initiated. Both individuals did not survive and did also not receive the appropriate treatment since high carbohydrate supplementation was given based on a misdiagnosis of a urea cycle disorder or tyrosinemia. In all other cases with appropriate treatment, the course of the disease was only mild.

Our cases further highlight the broad phenotypic spectrum, showing intermediate types in addition to the three classically defined subtypes (neonatal intrahepatic cholestasis caused by citrin deficiency (‘NICCD’), failure to thrive and dyslipidaemia caused by citrin deficiency (‘FTTDCD’), adult-onset recurrent hyperammonaemia with neuropsychiatric symptoms in citrullinemia type II (‘CTLN2′)) and should prompt genetic evaluation early in the course of suggestive cases. 

### 4.5. Limitations

Taking into account the rarity of these disorders, we are able to add a considerable number of new subjects and expand the scarce experience in this field. However, the small number of subjects reported and the retrospective study design are obvious weaknesses of the study. The KDs were not uniform and blood BHB levels were not available in all individuals, though, at least, all received a classical KD. Still, the lack of a standardised KD (e.g., by a target BHB-level range) limits the interindividual comparability of the effect. In some children, an increase in the cKD ratio improved the effect and we cannot exclude that this might have led to a greater benefit also in others.

## 5. Conclusions

The MAS/MPC1- defects are a group of rare diseases with either a neurological phenotype with early onset epilepsy, muscle hypotonia and a severe global developmental impairment or a hepatopathy (citrin deficiency). The course is progressive and severe if left untreated, often leading to early death. The majority of individuals benefited from the treatment with KD (or fat enrichment/carbohydrate restriction in case of hepatopathy/citrin deficiency). This study—once again—highlights the need for (early genetic) diagnosis to aid pathomechanism-based treatment. It further underlines that not only paediatric but especially adult individuals with complex disorders should be referred to specialized centres with expertise in the diagnosis and treatment of rare diseases.

## Figures and Tables

**Figure 1 nutrients-14-03605-f001:**
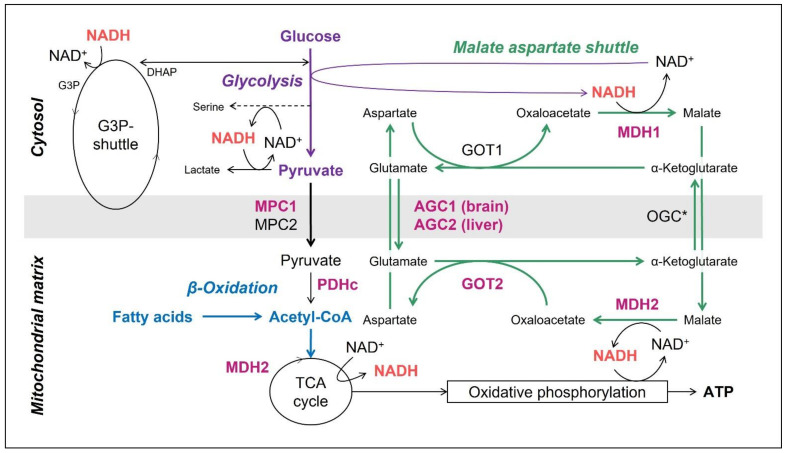
**Malate aspartate shuttle (MAS) and the mitochondrial pyruvate carrier (MPC).** The MAS is mainly responsible for the transfer of NADH across the inner mitochondrial membrane. MPC transport the end product of glycolysis—pyruvate—to the mitochondrial matrix, where it enters TCA cycle and finally oxidative phosphorylation. Fatty acids—from *ketogenic diet*—enter β-oxidation and thus bypass MAS and MPC. Proteins associated with human disease are written in bold pink. Glycolysis is inked in purple, MAS in green and β-oxidation (the bypass on KD) in blue. Of note, MDH2 takes part within the MAS and the TCA cycle. * Somatic variants in OGC have been related to pheochromocytoma. Abbreviations: NAD = nicotine amid adenine dinucleotide (NAD+ oxidated form, NADH reduced form), AGC1 = aspartate glutamate carrier, AGC2 = citrin, OGC1 = oxaloglutarate carrier, MDH1/2 = malate dehydrogenase (cytosolic/mitochondrial isoenzymes), GOT1/2 = aspartate aminotransferase (cytosolic/mitochondrial isoenzymes), MPC1/2 = mitochondrial pyruvate carrier 1/2 complex. G3P = glycerol-3-phosphate, DHAP = dihydroxyacetone phosphate, TCA = tricarboxylic acid.

**Figure 2 nutrients-14-03605-f002:**
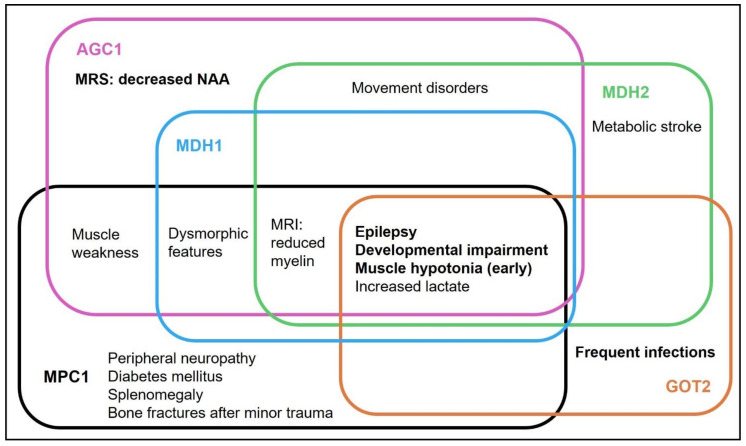
**Phenotypic overlap between different types of neurological malate aspartate shuttle (MAS) and mitochondrial pyruvate carrier (MPC) defects.** Common features are written in **bold**. AGC1 = aspartate glutamate carrier 1/2, MDH1/2 = malate dehydrogenase (cytosolic/mitochondrial isoenzymes), GOT2 = aspartate aminotransferase, MPC1 = mitochondrial pyruvate carrier 1 complex. NAA = N-acetyl-aspartate.

**Figure 3 nutrients-14-03605-f003:**
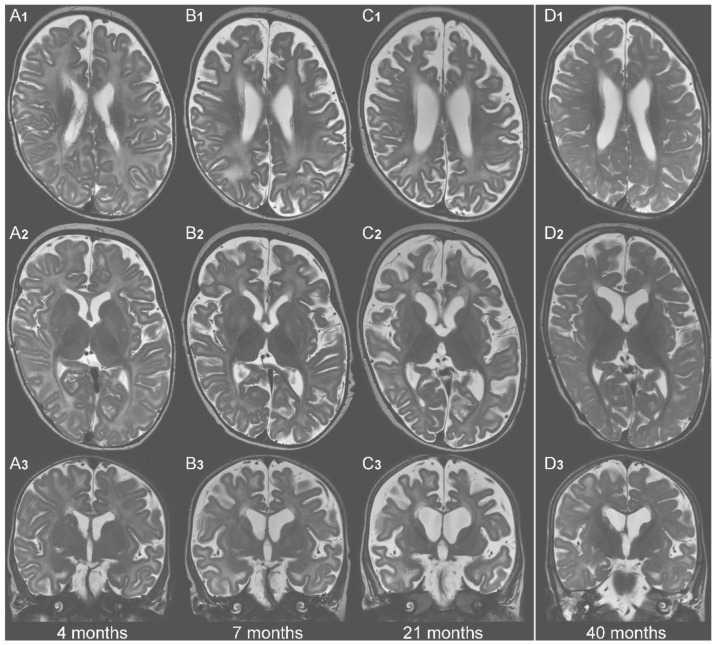
**MRI course AGC1-1.** Selected T2-weighted images from subject AGC1-1 at identical ana- tomical planes to allow comparison at four different time points: (**A1**–**D1**) at axial level through the upper part of the body of the lateral ventricles; (**A2**–**D2**) at axial level of the foramen of Monroe; (**A3**–**D3**) at coronal plane through the foramen of Monroe. (**A1**–**A3**) MRI at 4 months showing enlarged size of lateral ventricles and extracerebral space as well as inappropriate myelination. The MRI (**B1**–**B3**) at 7 months is clearly abnormal with evidence of cerebral volume loss and widespread white matter signal alteration, compatible with a secondary hypomyelination. The findings are subsequently progressive, with impressive volume loss and extensive white matter changes at 21 months (**C1**–**C3**). The MRI (**D1**–**D3**) at age 40 months, after 18 months on classical KD (cKD), showed clear improvement. The size of the ventricles and the extracerebral space was regressive, myelination in the cerebral white matter advancing in general, and in the optic radiation, the corpus callosum and the subcortical (gyral) areas, in particular.

**Figure 4 nutrients-14-03605-f004:**
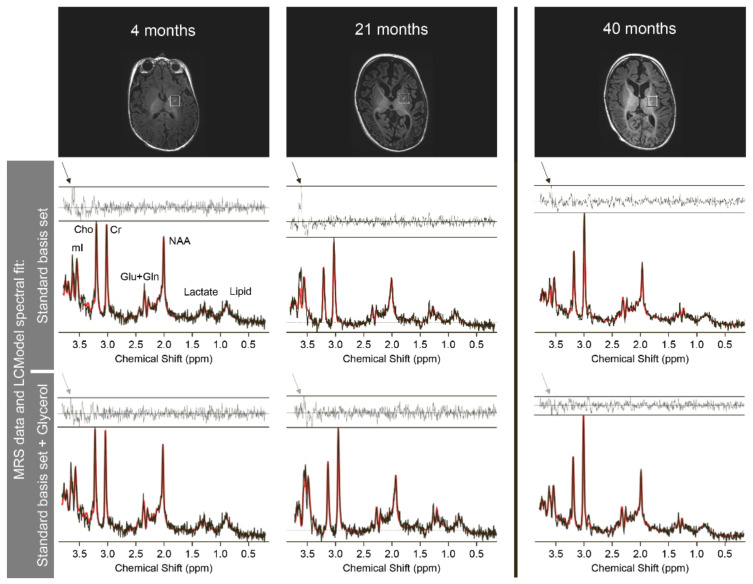
**MRS course AGC1-1.** Basal ganglia voxel positions and corresponding MR spectra acquired at an age of 3 months (left) and 1 year 9 months (middle), before introduction of classical KD (cKD), and at 3 years 4 months (right), after 18 months on the cKD. MRS data were analysed with LCModel, a fully automated spectral fitting package, which models each in-vivo spectrum as the linear combination of known basis spectra from a standard set of metabolites (see Appendix A for more information). For each spectrum, the raw MRS data are depicted in black and the LCModel fit is overlaid in red. The residuals between the fit and the data are plotted above each spectrum. Using the standard basis set, a peak is seen in the residuals at 3.6–3.65 ppm (black arrows), indicating the presence of a metabolite in the spectrum, which is not included in the basis set. This peak appears most prominent in the spectrum acquired at 1 year 9 months (middle row, middle column), and diminishes in size following introduction of the cKD (middle row, right column). After including glycerol in the LCModel basis set (bottom row), this residual peak decreases in size (grey arrows), and the spectral fit in the area of the spectrum around 3.6 ppm improves, suggesting that glycerol may contribute to this unknown signal. (mI: myo-inositol, Cho: Choline, Cr: Creatine, Glu + Gln: Glutamate + Glutamine, NAA: N-acetyl-aspartate).

**Figure 5 nutrients-14-03605-f005:**
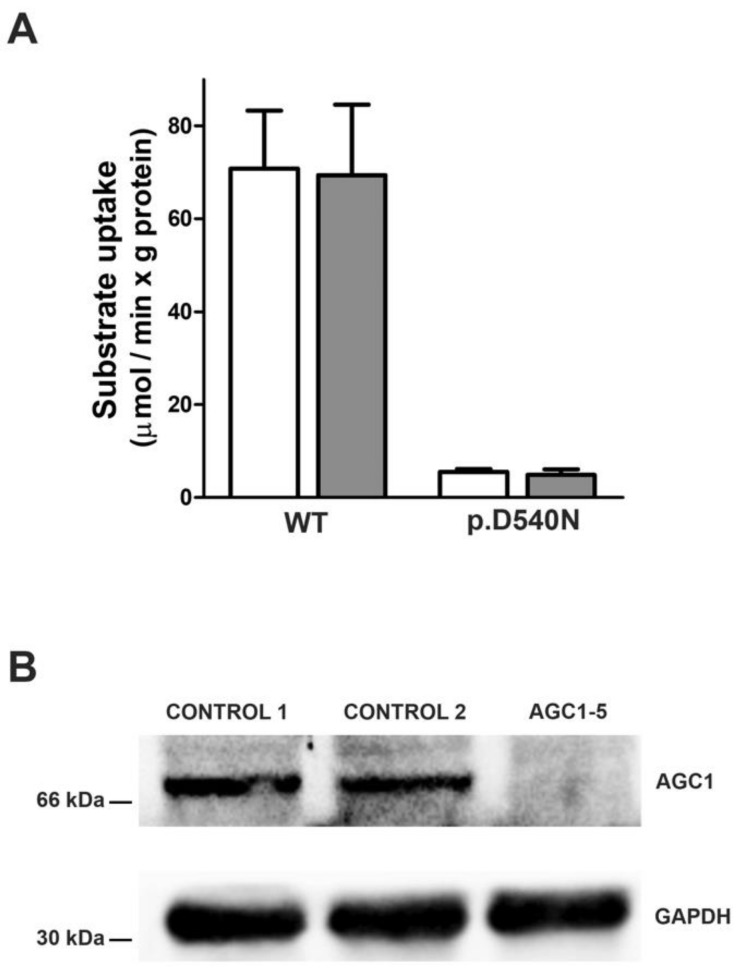
**Functional validation of AGC1 variants in AGC1-2/3 and AGC1-5.** (**A**) Transport assays of wild-type and p.D540N AGC1 variant in liposomes. Wild type (WT) and the p.D540N AGC1 variant of AGC1-2 and AGC1-3 patients were overexpressed in *Escherichia coli*, purified and reconstituted in liposomes, as previously described [40]. The uptake rate of ^14^C-glutamate (white bars) or ^14^C-aspartate (grey bars) was measured by adding 1 mM of radiolabelled glutamate or aspartate to liposomes reconstituted with purified WT or p.D540N AGC1 and containing 20 mM of glutamate. Transport reaction was terminated after 1 min by adding 20mM of pyridoxal 5′-phosphate and 20 mM batophenathroline. The means and SDs from three independent experiments are shown. (**B**) Expression analysis of AGC1 in fibroblasts from unrelated healthy controls and AGC1-5 proband. 100 μg of fibroblasts cultured at 37 °C in a humidified atmosphere with 5% CO2 in high glucose DMEM (D6546 SIGMA) supplemented with 10% foetal bovine serum and glutamine 2 mM, were harvested at passage #3 and lysed for western blot analysis with an antibody against AGC1. Densitometry analysis revealed the absence of AGC1 in AGC1-5 cells in comparison with two unrelated healthy controls showing similar AGC1 expression. An antibody against GAPDH was used for protein expression normalisation. The full western blot analysis is provided in Appendix A).

**Table 1 nutrients-14-03605-t001:** **a–d**: **Phenotype of neurological MAS without/before ketogenic diet**. **Legend 1. a–d:** in **bold** are the patients that received > 1 month ketogenic diet or serine & pyridoxine, respectively. n/r = not reported, wg = weeks of gestation, ↓ = reduced, ↑ = increased, d = day(s), w = week(s), m = month(s), p = percentile, prim = primary, sec = secondary, regr = regression, prog = progressive, neg = negative, gen = generalized, FU = follow up, AT= at term, PT = preterm, CS = cesarean section, preg = pregnancy, neo = neonatal, NICU = neonatal intensive care unit, NRDS = neonatal respiratory distress syndrome, DI: developmental impairment, SO = seizure onset, SE = status epilepticus, GOR = gastro-oesophageal reflux, FTT = failure to thrive, LFT = liver function test(s), AB = antibody, † = deceased, met = metabolic, resp = respiratory, dysm = dysmorphic, ASM = anti-seizure-medication, BRV = brivaracetam, esliCBZ = eslicarbazepine, LAC = lacosamide, LEV = levetiracetam, LTG = lamotrigine, CBZ = carbamazepine, OxCBZ = oxcarbazepine, Pb = phenobarbitone, CLB = clobazam, CLN = Clonazepam, VGB = vigabatrin, VPA = valproate, Pred = prednisolone, RUF = rufinamide, STM = sulthiame, STP = stiripentol, TPM = topiramate, ZNS = zonisamide.

**a: Phenotype of individuals with AGC1-defect without/before ketogenic diet**														
**#, Sex**	**Pregnancy & Birth**	**Age of Onset (m); Presenting Findings**	**Onset of Seizures (m)**	**Last FU (y, m)**	**Start KD (y, m)**	**Development At Start KD (Age If No/Before KD)**	**Developmental Course**	**Muscle Tone at Start KD (Last FU If No KD)**	**Movement Disorder**	**Seizure Semiology**	**ASM Before KD**	**Microcephaly**	**Other Findings Before Start KD**	**Reference**
**AGC1-1, m**	AT, polyhydr- amnios	birth; NRDS, muscle tone ↓	4	4 y 3 m	1 y 9 m	no social interaction, movements ↓, tube fed	sec DI upon SO	↓	choreo- athetotic	focal clonic; (multi)focal with apnea; tonic eye deviation & laughter, +/− motor	LEV, OxCBZ, Pb, LTG, CLB, VGB, TPM, Pred, STM, RUF, STP	yes, sec	GOR/vomiting, FTT, feeding difficulties, LFT ↑; non-epileptic apnea, dysm, muscle weakness, tongue fibrillation	this paper
**AGC1-2, m**	AT	5 m; seizures	5	7 y 8 m (6 m after stopp KD)	5 y 8 m	non-verbal, spoon-fed, no sitting	sec DI upon SO	↓	no	focal clonic	LEV	yes	FTT	this paper
**AGC1-3, f**	AT, CS	5 m; muscle tone ↓	7	3 y 11 m (2 w after stopp KD)	1 y 4 m	no visual fixation	sec DI, 3 m before SO	↓	no	focal with starring, motor arrest, cyanosis	LEV	no, p3-10	growth retardation	this paper
AGC1-4, f	AT; foetal move-ments ↓	birth; lethargic, muscle tone ↓	5	7 y (off)	3 y (1 m KD)	non-verbal, no motor milestones, tube fed	sec DI, 1 m after SO	↓	dystonic- spastic (7y)	n/r	Pb, LEV, STM, OxCBZ. LAC, (7y): OxCBZ	yes, sec	GOR/vomiting, FTT	this paper
**AGC1-5, m**	AT	birth; lethargic, breast-feeding difficulties	7	4 y 6 m	2.5	no sitting, no crawling, non-verbal, interaction ↓, eye contact ↓, tube fed	prim DI, worse- ning upon SO	↓	hypo-tonic- dys-kinetic	tonic	LEV, TPM, esliCBZ	yes, sec	feeding difficulties	this paper
**AGC1-6, m**	AT	6 m; seizures	6	1 y 5 m	1 y 2 m	babbeling	global DI	↓	no	focal with apnea	LEV, TPM	n/r	n/r	this paper
**AGC1-7, f**	preg normal	5 m; DI	7	7 y 8 m	6	no head control, rolling, grasping, eye contact ↓, smiling response	sec DI, stag- nation after SO	↓	n/r	apneas, tonic, focal clonic, myoclonic, lip smacking, chewing, laughter	LEV, CBZ, OxCBZ	no	n/r	[16,23]
AGC1-8, f	PT (36 wg)	birth; hypogly-cemia (7 d NICU), muscle tone ↓	10	6 y 8 m	n/a	non-verbal, smiling, sitting assisted, drooling	sec DI, regr after SE	↓	no	focal clonic, gen tonic-clonic	TPM, Pb, ZNS	yes, sec	dysm, short stature	[24]
AGC1-9, m	PT, CS (33 wg)	birth; polyhydr- amnios	10	1 y 1 m	n/a	bearing weight, sitting, rolling over, smiling	n/r	↓	no	focal, clonic, sec bilateral	LEV	no	dysm, jejunal atresia	[24]
**AGC1-10, m**	AT, preg normal	7 m; febrile seizures	7	>2 y 1 m (4 m KD)	> 1 y 9 m	sitting assisted, monosyllables	sec DI upon SO	↓	n/r	tonic-clonic, myoclonic	n/r	n/r	not dysm	[17]
AGC1-11, n/r	CS	birth	8	n/r	n/a	n/r	n/r	↓	n/r	n/r	n/r	n/r	ptosis	[25]
AGC1-12, m	AT, delayed transition	birth; oxy-gen (8 d)	3	11 y	n/a	sitting with support, smiling, vocalization, understanding some language, tube fed	prim DI, regr upon SO	↑	dystonic-spastic (11y)	focal eye/head deviation, control at 3 y. At 12y relapse, starring spells	OxCBZ, LEV, TPM, Pb	yes	GOR, ear infections, blepharitis, osteopenia, scoliosis, optic neuropathy, CVI, short stature	[26]
AGC1-13, f	n/r	1 m; seizures	1	7 y (†)	n/a	severe DI, non-verbal	n/r	↑	n/r	n/r	n/r	n/r	subtle dysm	[27]
AGC1-14, f	n/r	3 m; n/r	n/r	1 y 4 m	n/a	n/r	n/r	n/r	n/r	n/r	n/r	n/r	optic atrophy	[28]
**b: Phenotype of individuals with MDH1/2 defect without/before ketogenic diet.**														
**#, Sex**	**Pregnancy & Birth**	**Age of Onset (m); Presenting Findings**	**Onset of Seizures (m)**	**Last FU (y, m)**	**Start KD (y, m)**	**Development At Start KD (Age If No/Before KD)**	**Developmental Course**	**Muscle Tone at Start KD (Last FU If No KD)**	**Movement Disorder**	**Seizure Semiology**	**ASM Before KD**	**Microcephaly**	**Other Findings Before Start KD**	**Reference**
MDH1-1, m	PT (32 wg), preg normal	3 m; micro-cephaly, dysm	13	2 y 6 m	n/a	DI, (13 m: rolling over, some babbling)	n/r	↑	n/r	hypsarrhyth-mia (infantile spasms)	TPM, CLN	yes, prog	growth retardation, strabism	[29]
MDH1-2, f	n/r	n/r	n/r	4 y	n/a	DI, (walking 3 y, a few words 4 y)	n/r	↑	n/r	n/r	n/r	yes	dysm	[29]
**MDH2-1, m**	AT, preg normal	5 m; muscle tone ↓, no head control	7	4½ y (5 y alive)	3 y	DI, 18 m sitting, crawling, good eye contact, averbal, tube fed	n/r	↓	dys-kinetic / dystonic	myoclonic	n/r	yes	FTT, constipation, strabism	[30]
**MDH2-2, m**	preg normal	birth; constipation	2	1 y 8 m (†)	1 y 6 m	DI, 1y: no sitting, no crawling; good eye contact, babbling	n/r	↓	no	gen tonic-clonic, spasms	n/r	no	strabism	[30]
**MDH2-3, m**	AT	birth; muscle tone ↓, macro-cephaly/-somia, two add. nipples	(?)	7 ½ y (12 y alive)	3 y	no crawling; no good eye contact; no language, tube fed	n/r	↓	dystonic	myoclonic, gen tonic	n/r	no	FTT, loss of vision, von Willebrand disease, congenital cystic adenomatoid malformation	[30]
MDH2-4, f	AT, preg normal	1st m of life; febrile seizures	1st m	4 y	3 y (Tri- hepta-noin)	DI	n/r	↓	dys-kinetic / choreatic	gen, absences	LTG, LEV	yes	1st year: FTT. 18 m met stroke; episodes of met decompensation	[31]
**c: Phenotype of individuals with GOT2 defect without ketogenic diet.**														
**#, Sex**	**Pregnancy & Birth**	**Age of Onset (m); Presenting Findings**	**Onset of Seizures (m)**	**Last FU (y, m)**	**Start KD (y, m)**	**Development At Start KD (Age If No/Before KD)**	**Developmental Course**	**Muscle Tone at Start KD (Last FU If No KD)**	**Movement Disorder**	**Seizure Semiology**	**ASM Before KD**	**Microcephaly**	**Other Findings Before Start KD**	**Reference**
GOT2-1, m	AT, CS	1 m; DI, muscle tone ↓, abdominal spasms, feeding difficulties	9	7 y 10 m	n/a	profound DI, no words (6 m: no head control, no sitting, no visual fixation)	n/r	↑	n/r	upward gaze, clonic seizures (left/right)	VPA, Pb, LEV, CBZ, LTG, TPM	yes	frequent infections, sleep disturbance	[32]
GOT2-2, f	PT (32 wg)	birth; DI, drooling, feeding difficulties	7	10 y	n/a	severe DI, <10 words, following objects, smiling	n/r	↑	n/r	myoclonic, gen tonic-clonic, tonic	LTG, VPA	yes	frequent infections, acrocyanosis and chillblains	[32]
GOT2-3, f	AT	birth; DI, drooling, feeding difficulties	6	8 y	n/a	severe DI, sitting, using hands, following objects, smiling, vocalize	n/r	↑	n/r	myoclonic, gen tonic-clonic	VPA, LTG, LEV	yes	frequent infections, acrocyanosis and chillblains	[32]
GOT2-4, m	AT	birth; feeding difficulties	4	4 y	n/a	profound DI, non-verbal, unable to follow objects, abnormal eye movements	n/r	↑	n/r	myoclonic, tonic (upward gaze)	VPA, LEV, VGB, CLN	yes	frequent infections	[32]
**d: Phenotype of individuals with MPC1-defect without/before KD.**														
**#, Sex**	**Pregnancy & Birth**	**Age of Onset (m); Presenting Findings**	**Onset of Seizures (m)**	**Last FU (y, m)**	**Start KD (y, m)**	**Development At Start KD (Age If No/Before KD)**	**Developmental Course**	**Muscle Tone at Start KD (Last FU If No KD)**	**Movement Disorder**	**Seizure Semiology**	**ASM Before KD**	**Microcephaly**	**Other Findings Before Start KD**	**Reference**
**MPC1-1, m**	AT, CS, breech, foetal move-ments ↓	birth; lactate ↑, resp distress	2.5	1 y 7 m	1 y 3 m	profound DI, no change of position, no grasping, eye contact/fixation ↓, non-verbal	prim DI (regr at 2 m)	↓	no	2 ½ m: myoclonia; 8 m: infantile spasms	LEV, CLB, OXC, LCS, VGB, RUF, LTG, BRV	yes, sec	muscle weakness	this paper
MPC1-2, m	AT	6 y (?); seizures	72 (?)	12 y	n/a	IQ 56	sec DI	normal	no	gen tonic clonic	VPA	yes	diabetes mellitus, splenomegaly, growth retardation (12 y p < 2), fractures	this paper
MPC1-3, f	AT	birth; truncal tone ↓	4	11 y	n/a	short sentences, counting to 5, walking	prim DI	↓	no	zyanotic, tonic, tonic-clonic	LTG	no	no	this paper
MPC1-4, m	AT	birth; truncal tone ↓	no	6 y	n/a	some words (sitting > 1 y, walking 4 y)	prim DI	↓	no	no seizures	n/a	no	no	this paper
**MPC1-5, f**	AT; induced delivery (mother)	birth; muscle tone ↓, dysm, hepato- megaly, resp distress	n/r	1 y 7 m	neo	1st m: rotatory nystagmus, poor visual contact, worsening over time	prim DI, sec regr	↓	n/r	n/r (no seizures?)	n/a	yes, prog	congenital heart defect, mild renal insufficiency, growth failure	[19,33,34]
MPC1-6, m	n/r	n/r	n/r	~20 y	n/r	mild DI (last FU)	primDI	↓	n/r	n/r (seizures)	n/r	n/r	n/r	[33,34]
MPC1-7, m	n/r	n/r	n/r	~17 y	n/r	severe DI (last FU)	n/r	n/r	n/r	n/r (no seizures?)	n/a	n/r	peripheral neuropathy, visual impairment	[33,34]
MPC1-8, f	n/r	n/r	n/r	~12 y	n/r	mild DI (last FU)	n/r	n/r	n/r	n/r (no seizures?)	n/a	n/r	peripheral neuropathy	[33,34]

**Table 2 nutrients-14-03605-t002:** **AGC2/citrin deficiency: phenotype and genetic findings**.

#, Sex	Country of Origin	Pregnancy, Delivery, Postnatal Course; Past History	Reason for Referral (Age)	Max. Blood Citrulline (μmol/L)	NH3 [Ref < 50 µmol/L]	Serum Galactose [Ref <20 mg/dl]	Other Lab Values	Hypo- gly- cemia	Ultrasound Liver	Treatment	Outcome/ Course
AGC2-1, m	Austria	unremark	hepatomegaly, -pathy (4 m)	104.8 (10–36)	normal	in urine ↑	liver transaminases ↑, blood threonine ↑, methionine ↑, ferritin ↑; prothrombin time ↓	no	mildly ↑ echogenicity of liver parenchyma	CH 40–45%, MCT 15–20%	lab normalised within weeks, thrives and develops age-adequate (2.4 y)
AGC2-2, m	Austria	n/a; epilepsy since adolescence	↑ NH3, seizures, vomiting (35 y)	571 (<50)	151	n/a	urinary arginino-succinic acid ↑	no	n/a	glu inf, prot restr, ammonia scavengers	death (35 y)
AGC2-3, f	Austria	SGA, T21, neonatal bilirubin ↑	NBS: citrulline ↑ (5 w)	940 (12.6–58)	normal	normal (NBS), ↑ to max 25 (blood)	threonine ↑, methionine ↑, tyrosine ↑, threonine/serine ratio ↑; hyperbilirubi-nemia/ cholestasis, liver transaminases ↑, disturbed clotting	no	mildly ↑ echogenicity of liver parenchyma	CH 40–45%, MCT 15–20%	lab normalised within weeks, thrives and develops age-adequate (4½ y)
AGC2-4, m	Austria	SGA	cholestatic jaundice (4 w), positive family history	523 (12.6–58)	normal	73.9 (<15)	unremark serum amino acids, liver transaminases, clotting parameters	no	unremark		lab normalised within 10 d, thrives and development age-adequate (2.2 y)
AGC2-5, m	Syria	neonatal hyperbili- rubinemia	suspected abdominal neoplasia (7 m)	28 (5–24)	79–182	n/a	tyrosine ↑, cholestasis, hemolytic anemia, absent succinylacetone in urine/dried blood spot	no	n/a	glu inf, prot restr, nitisinone	death within 7 d (7 m)
AGC2-6, f	Syria	unremark	positive family history, (diagnosis from cord blood on d 12)	764 (5–24) (45 d)	normal	normal (NBS), upon diagnosis ↑ (7542 (<800) μmol/L)	plasma threonine ↑	no	n/a		lab normalised within 7 d, thrives and development age-adequate (22 m)
AGC2-7, m	Syria	SGA	FTT (6 m)	465 (4–65)	normal	↑	cholestasis, triglycerides ↑	no	unremark	CH 26%, lactose and galactose free, MCT fat-enriched 61%	FTT improved, hepatopathy (5 y)
AGC2-8, f	Austria	uneventful	NBS: citrulli- ne ↑ (25 d)	860	103	4.8 → ↑ 90	cholestasis	no	mildly ↑ echogenicity of liver parenchyma	CH restr	normal growth and laboratory values (4½ y)

**Abbreviations in Table 2:** d = day(s), w = week(s), m = month(s), y = year(s), n/a = not available, unremark = unremarkable, SGA = small for gestational age, CH = carbohydrates, MCT = medium chain triglycerides, glu = glucose, prot = protein, inf = infusion, restr = restriction, NBS = new born screening, ↑ = increased.

**Table 3 nutrients-14-03605-t003:** **Genotype of all individuals**.

Patient ID	*Gene*	RefSeq	Variant	Predicted Change	HGMD	ACMG Classification	Reference
GOT2-1	*GOT2*	NM_002080.4	c.617_619delTTC	p.Leu209del	DM	PTH	[32]
GOT2-2, GOT-3	*GOT2*	NM_002080.4	c.784C > G	p.Arg262Gly	DM	PTH	[32]
GOT2-1	*GOT2*	NM_002080.4	c.1009C > G	p.Arg337Gly	DM	PTH	[32]
GOT2-4	*GOT2*	NM_002080.4	c.1097G > T	p.Gly366Val	DM	PTH	[32]
MDH1-1, MDH1-2	*MDH1*	NM_001199111.1	c.413C > T	p.Ala138Val	DM	LPTH	[29]
MDH2-3	*MDH2*	NM_005918.4	c.109G > A	p.Gly37Arg	DM	PTH	[30]
MDH2-1, MDH2-2, MDH2-3, MDH2-4	*MDH2*	NM_005918.4	c.398C > T	p.Pro133Leu	DM	PTH	[30]
MDH2-4	*MDH2*	NM_005918.4	c.445delinsACA	p.Pro149Hisfs*22	DM	PTH	[31]
MDH2-2	*MDH2*	NM_005918.4	c.596delG	p.Gly199Alafs*10	DM	PTH	[30]
MDH2-1	*MDH2*	NM_005918.4	c.620C > T	p.Pro207Leu	DM	LPTH	[30]
MPC1-2	*MPC1*	NM_016098.4	c.95C > G	p.Ala32Gly	not listed	LPTH	this paper
MPC1-1	*MPC1*	NM_016098.4	c.214A > G	p.Lys72Glu	not listed	LPTH	this paper
MPC1-3, MPC1-4, MPC1-6, MPC1-7, MPC1-8	*MPC1*	NM_016098.4	c.236T > A	p.Leu79His	DM	LPTH	[33,34]
MPC1-5	*MPC1*	NM_016098.4	c.289C > T	p.Arg97Trp	DM	LPTH	[33,34]
AGC1-14	*SLC25A12*	NM_003705.5	c.125G > C	p.Arg42Pro	not listed	LPTH	[28]
AGC1-5	*SLC25A12*	NM_003705.5	c.225del	p.Glu76Serfs*17	not listed	PTH	this paper
AGC1-13	*SLC25A12*	NM_003705.5	c.400C > T	p.Arg134*	not listed	PTH	[27]
AGC1-4	*SLC25A12*	NM_003705.5	c.810_811insA	p.Leu271Thrfs*9	not listed	PTH	this paper
AGC1-8, AGC1-9	*SLC25A12*	NM_003705.5	c.1058G > A	p.Arg353Gln	DM	LPTH	[24]
AGC1-12	*SLC25A12*	NM_003705.5	c.1295C > T	p.Ala432Val	DM	PTH	[26]
AGC1-10	*SLC25A12*	NM_003705.5	c.1331C > T	p.Thr444Ile	not listed	LPTH	[17]
AGC1-6, AGC1-11	*SLC25A12*	NM_003705.5	c.1335C > A	p.Asn445Lys	DM	LPTH	[25]
AGC1-12	*SLC25A12*	NM_003705.5	c.1447-2_1447-1delAG	p.?	not listed	PTH	[26]
AGC1-1	*SLC25A12*	NM_003705.5	c.1586-?_1835 + ?del (deletion exon 16 and 17, exact break-point not determined)	p.?	not listed	PTH	this paper
AGC1-2, AGC1-3	*SLC25A12*	NM_003705.5	c.1618G > A	p.Asp540Asn	not listed	LPTH	this paper
AGC1-5	*SLC25A12*	NM_003705.5	c.1747C > A	p.Arg583Arg	not listed	LPTH	this paper
AGC1-7	*SLC25A12*	NM_003705.5	c.1769A > G	p.Gln590Arg	DM	LPTH	[16,23]
AGC2-7	*SLC25A13*	NM_014251.3	c.173_174delTG	p.Val58Glyfs*24	not listed	PTH	this paper
AGC2-1, AGC2-8	*SLC25A13*	NM_014251.3	c.848 + 1G > T	p.?	not listed	PTH	this paper
AGC2-2, AGC2-3, AGC2-4	*SLC25A13*	NM_014251.3	c.1078C > T	p.Arg360*	DM	PTH	[35]
AGC2-1	*SLC25A13*	NM_014251.3	c.1173T > G	p.Tyr391*	not listed	LPTH	this paper
AGC2-2, AGC2-8	*SLC25A13*	NM_014251.3	c.1307_1308delinsAA	p.Gly436Glu	not listed	LPTH	this paper
AGC2-5, AGC2-6	*SLC25A13*	NM_014251.3	c.1629dup	p.Ile544Tyrfs*24	not listed	PTH	this paper
AGC2-7	*SLC25A13*	NM_014251.3	c.1813C > T	p.Arg605*	DM	PTH	[36]

**Abbreviations:** HGMD = Human Gene Mutation Database, ACMG = American College of Medical Genetics and Genomics, DM = disease causing mutation, PTH = pathogenic, LPTH = likely pathogenic.

**Table 4 nutrients-14-03605-t004:** **Course on ketogenic diet in individuals with neuronal MAS/MPC1 defects**.

#, sex	Start (y)	Duration (m)	Last FU (y)	Composition of KD	BHB (mmol/L)	General Effect	Effect on Epilepsy	Effect on EEG	Effect on Motor Abilities	Muscle Tone & Head Control	Effect on Communication	Effect on Thriving/Feeding	Effect on MRI	Effect on MRS	Effect on Lab
**AGC1-1, m**	1.75	30	4.25	3:1 → 4:1	2.5–3.5	++	++	+	++	↑	+	+	++	+	LFT ↓ L (n)
**AGC1-2, m**	5.67	18	7.67	2:1	n/r	+	(+)	+	0	0	0	0	n/a	n/a	n/r
**AGC1-3, m**	1.3	31	3.92	2:1	0.4–4.0	+	+	+	++	↑	+	0	n/a	n/a	L ↓
**AGC1-4, m**	3	1	3.08	3:1	n/r	0	0	n/r	0	0	0	0	n/a	n/a	n/r
**AGC1-5, m**	2.5	12	4.5	3:1	3–4	++	++	+	++	↑	+	+	n/a	n/a	n/r
**AGC1-6, m**	1.17	3	1.42	3:1	n/r	+	++	n/a	+	↑	n/r	n/r	n/a	n/a	n/r
AGC1-7, f	6	20	7.67	3:1→4:1	5.5–6.8	++	++	+	++	↑	+	n/r	++	+	0
AGC1-10, m	1.75	4	n/r	4:1	n/r	++	++	n/r	n/r	↑	n/r	n/r	n/a	n/a	n/r
MDH2-1, m	3	18	4.5	n/r	n/r	+	+	n/r	n/r	n/r	n/r	n/r	n/r	n/r	n/r
MDH2-2, m	1.5	2	1.66	n/r	n/r	+	+	n/r	n/r	n/r	n/r	n/r	n/r	n/r	n/r
MDH2-3, m	3	54	7.5	n/r	n/r	+	?	n/r	n/r	n/r	n/r	n/r	n/r	n/r	n/r
**MPC1-1, m**	1.25	36	4.25	2.5:1→4:1	2–3	+	++	++	+	↑	+	n/r	n/a	n/a	L ↓
MPC1-5, f	neo	19	†	n/r	<1.5; 0.3–3.7 (>16 m)	0	n/r	n/r	-	0	-	n/r	n/a	n/a	0

**Abbreviations:** + positive effect, ++ striking positive effect; —negative effect, -- striking negative effect, 0 = no effect, ↓ = decrease, ↑ = increase, FU = follow up, m = month(s), n/r = not reported, n/a= not applicable, neo = neonatal, † = deceased, LFT = liver function test, L = lactate, n = normal.

**Table 5 nutrients-14-03605-t005:** **MRS in AGC1-1 before and on ketogenic diet**.

	No KD			KD	No KD			KD
Age	4 m	7 m	21 m	40 m	4 m	7 m	21 m	40 m
	**white matter concentrations**				**basal ganglia concentrations**			
Lac (mM)	0.73	2.01	2.53	0.98	0.73	0	2.57	1.54
mI (mM)	6.42	6.14	10.1	4.66	4.7	10.67	6.81	4.84
Asp (mM) *	0	0	0	0.68	0	0.16	0.21	1.11
NAA (mM)	3.67	2.35	2.07	3.08	3.63	3.47	3.19	3.42
	**white matter NAA/Cr-ratio**				**basal ganglia NAA/Cr-ratio**			
NAA/Cr	0.45	0.46	0.32	0.38	0.7	0.34	0.37	0.4
	**white matter concentrations**				**basal ganglia concentrations**			
Glyc (aU) *	5.73	33.44	20.92	6.4	13.04	20.64	23.88	0
Abbreviations: Lac = lactate, mI = myo-inositol, Asp = aspartate, NAA= N-acetyl-aspartate, Cr = creatinine, Glyc = glycerol, aU = arbitrary unit (scaling factor based on a phantom of known glycerol-3-phosphate concentration), m = month(s) * = no reference values available. Means and SD of mI/NAA-concentrations are age-dependent and were taken from Pouwels et al. [37] (cf. their tables 3 & 6)								
Color codes:	normal (for Lac <2mM; for mI: within mean *±* 2.5 SD [37])							
	increased (for Lac >2mM; for mI: above mean + 2.5 SD [37])							
	reduced (for NAA: below mean−2.5 SD [37]; NAA/Cr: <1)							

## Data Availability

All data are presented in this manuscript.

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
