# Peer review of "Ketogenic Diet Treatment of Defects in the Mitochondrial Malate Aspartate Shuttle and Pyruvate Carrier"

_nutrients, 2022, doi:10.3390/nu14173605_

Round 1

Reviewer 1 Report

Bölsterli et al present a comprehensive analysis of clinical presentation in a large (40 subjects) cohort of patients with alterations in either the malate-aspartate shuttle proteins or the mitochondrial pyruvate carrier and evaluate the effect of Ketogenic diet (KD) when used as treatment.  MAS/MPC associated mutations are rare and therefore the number of patients identified so far was scarce, providing limited knowledge on symptomology and efficacy of treatments. In this article, the authors confirm, update and extend previously described manifestations of MAS/MPC-related deficits and the benefits of dietary therapy with KD to a large cohort which includes previous and novel subjects. The cohort was found through literature search and almost half are novel patients recruited from a collaborative research. The compiled data are clearly summarized in tables and properly discussed in the text, highlighting the similarities in the presentation associated to the different genes and the beneficial outcome of increasing the ratio fat to non-fat nutrients, stressing the importance of early diagnosis. Also, they point to AGC2/citrin-deficiency as a complex disorder to be taken into account when diagnosing urea cycle defects.   

Comments:

-- The rational for decreased Serine biosynthesis is the requirement of oxidized cofactor NAD+, which is expected to be low in MAS deficiency. However, this applies to neurons and not astrocytes, as discussed elsewhere based on absent/low levels of AGC1 in glial cells. Interestingly, in brain Serine is synthesized by astrocytes, not neurons (PMID: 32798642). Another possibility to explain the reduced Serine would be increased transformation to pyruvate, as suggested by Pardo et al 2011 (ref 42). This alternative should be included in the discussion.

-- Line 95. “the MAS plays an important role in mitochondrial respiration”. This affirmation is not completely correct.  Mitochondrial respiration is considered the production of energy in the oxidative phosphorylation, typically using NADH (or FADH2) generated in TCA or b-oxidation, and also the NADH transferred to the matrix by the shuttles. Mitochondrial respiration on substrates that do not require maintenance of NAD/NADH ratio in the cytosol can take place in absence of MAS. The concept will be more accurately given if  “when glucose or lactate are the substrates” or similar is added. Similarly in Legend to Figure 1, “ which is necessary for mitochondrial respiration” should be eliminated.

-- Line 131. Please correct typo “par” to “part”.

-- Line 314. “Of note, none had shown hypoglycemia”. The significance of this statement is somewhat lost without presenting first the clinical characteristics of AGC2 deficiency, which typically include hypoglycemia (Saheki & Song 2017, ref 11). Readers would benefit from the information if provided.

Line 377.  See comment for line 95.

- Line 385: Please correct statement “AGC2 is the isoenzyme outside the brain”. AGC2 is the main isoform expressed in liver, not the only isoenzyme outside the brain. In fact, AGC1 is highly expressed also in skeletal muscle, heart, kidney and pancreatic cells.  

Line 401. Please correct typo “rule” to “role”.

Line 436.  The hypothesis of reduced Asp leading to low levels of NAA and therefore myelin was first proposed in the AGC1- KO mouse (Ref 41 , Jalil et al), which should be cited here as well.

-- Table 5. Please provide myo-Inositol concentrations range.

--Full blot of western blot in figure 5 should be provided as supplementary material.

-- Supplemental methods:  Information on how and for how long the fibroblasts were maintained in culture before preparation of cell extracts should be provided. Please correct units to show the adequate symbol instead of a square.

-- Supplemental Figure 1:  The methodology for evaluating substrate oxidation and respiratory complex activities needs to be provided, as well as the number of replicates shown in the figures and the raw value for the lowest normal used in normalization. Furthermore, in the right panel complex V is seems to be even higher than PDH (when bars are compared) but this is not discussed.

-- Formatting of references (brackets) through the text is missing.

-- Please provide abbreviation definitions as legend for the terms used in tables 1-4.

Reviewer 2 Report

This retrospective analysis and systemic review of the role of a ketogenic diet for genetic mutations linked to defects in the mitochondrial malate aspartate shuttle and pyruvate carrier is very well written and very informative. Collating all of these data in one manuscript (clinical story, signs and symptoms, genetic data, MRI and MRS and dietary interventions) is comprehensive reference for pediatric geneticists that manage patients with here disease. teh addition of the biochemical discussion also adds value to this manuscript. I only have minor comments:

1. In the materials and methods section the authors should state that this is predominantly a retrospective review and also state if data from some subjects were collated in a more prospective manner. Currently, this is not well defined.

2. Table 4 should have a key to define the symbols.

3. The limitations section should state that the ketogenic diets were not uniform and the question is whether the discussion should speculate on the significance of this?
